# The gene-regulating proteins NONO and SFPQ assemble into ordered filaments

Tim Rasmussen [1,2], Jannik Küspert[3], Lars Schönemann[1], Dietmar Geiger[3] ✉ & Bettina Böttcher [1,2] ✉

Proteins of the Drosophila behaviour/human splicing (DBHS) family are involved in many aspects of gene regulation and maintenance like transcription, splicing and DNA repair. DBHS proteins form obligate homo- and heterodimers through interactions within a globular domain and can further dynamically oligomerise through α-helical coiled-coils, which is crucial for many functions. While the atomic structures of the dimers are established, the arrangement in higher oligomers is unknown. Here we present the structure of a filamentous NONO/SFPQ heterooligomer resolved by cryo-EM. The filaments form a double helix which is stabilized by an interdigitating network of coiled-coil interactions.

Proteins of the Drosophila behaviour/human splicing (DBHS) family are highly conserved in animals and are involved in many gene regulation processes at DNA and RNA level[1,2]. Humans possess three members of the DBHS family: Non-POU domain containing octamer binding protein (NONO; also abbreviated p54[nrb]), splicing factor proline/glutamine rich (SFPQ; also called polypyrimidine-tract binding protein splicing factor, PSF) and paraspeckle protein component 1 (PSPC1). NONO and SFPQ interact with the C-terminal domain of RNA polymerase II from initiation to termination[3]. After transcription, NONO and SFPQ have been implicated in RNA splicing and polyadenylation[4–6]. DBHS proteins together with the long non-coding RNA NEAT1 also seed and develop subnuclear bodies called paraspeckles[7]. These paraspeckles play a role in gene regulation by retaining RNA within the nucleus[8,9]. In addition, paraspeckles sequester free SFPQ from the nucleoplasm, which influences gene expression because SFPQ can bind to DNA and serves as a repressor or activator of certain genes[10,11]. Considering the multiple roles of DBHS proteins in gene regulation, it is not surprising that they are involved in higher functions of the mammalian cell, like development, cell cycle and circadian rhythm. Consequently, malfunction leads to severe health problems as for example intellectual disabilities[12] and cancer[13].

The molecular organisation of DBHS proteins and their common domains have been revealed by sequence alignments and X-ray structures of homo- and heterodimers[14–22]. The DBHS core consists of two RNA recognition motifs (RRMs), together with a NonA/paraspeckle domain (NOPS) and a coiled-coil domain (Fig. 1a). N- and C-terminal extensions differ between NONO, SFPQ, and PSPC1 and include disordered regions and nuclear localisation signals. In addition, SFPQ possesses an uncharacterised DNA-binding domain (DBD), which is located N-terminally to the DBHS core. Long α-helices extending from the globular domain showed interaction with other dimer units in crystals[22,23] which inspired models of oligomerisation[14,24,25]. SFPQ requires a specific coiled-coil interaction motif (Fig. 1a, region 1) for transcriptional regulation, paraspeckle formation and DNA binding[23], highlighting the functional role of oligomerisation.

Despite many functional studies on DBHS proteins, there is only a limited understanding of the molecular mechanisms behind their cellular roles. As DBHS proteins lack catalytic activity, their contribution to gene regulation could be the provision of a common scaffold and binding platform for protein, RNA and DNA. The molecular structure of the DBHS dimers, though well characterised, fails to fully account for the higher oligomerisation of these dimers, which is crucial to many of their functions. To fill this gap, we present the polymeric DBHS structure consisting of flexible helical filaments.

## Results and Discussion

The native full-length NONO/SFPQ was purified[26] from Chinese hamster ovary cells (CHO; *Cricetulus griseus*; Supplementary Fig. 1) which are highly conserved to the human homologues (Supplementary Fig. 2-4). Electron micrographs showed a concentration dependent filament formation (Fig. 1b–d). The filaments were inherently flexible with a varying local curvature (Fig. 1d) but had a constant diameter of 15 nm, also evident from 2D-class averages (Fig. 1e, Supplementary Fig. 5). Overall, a resolution of 3.9 Å was obtained with local variations (Table 1, Supplementary Fig. 6). The protruding globular domains were less well resolved than the interior of the filament (Supplementary Fig. 7). The DBD of SFPQ and the disordered regions are not visible in our structure. The highest resolution of 3.3 Å was obtained in local refinements of 4 dimers on one strand (Table 1, Supplementary Fig. 6-8). 3D-classification revealed the flexibility and the relative movement of the protruding globular domains (Supplementary Movie 1).

[1]University of Würzburg, Rudolf Virchow Centre, Würzburg, Germany. [2]University of Würzburg, Biocentre, Chair of Biochemistry II, Würzburg, Germany. [3]University of Würzburg, Julius-von-Sachs-Institute for Biosciences, Department of Molecular Plant Physiology and Biophysics, Würzburg, Germany. ✉e-mail: dietmar.geiger@uni-wuerzburg.de; Bettina.boettcher@uni-wuerzburg.de

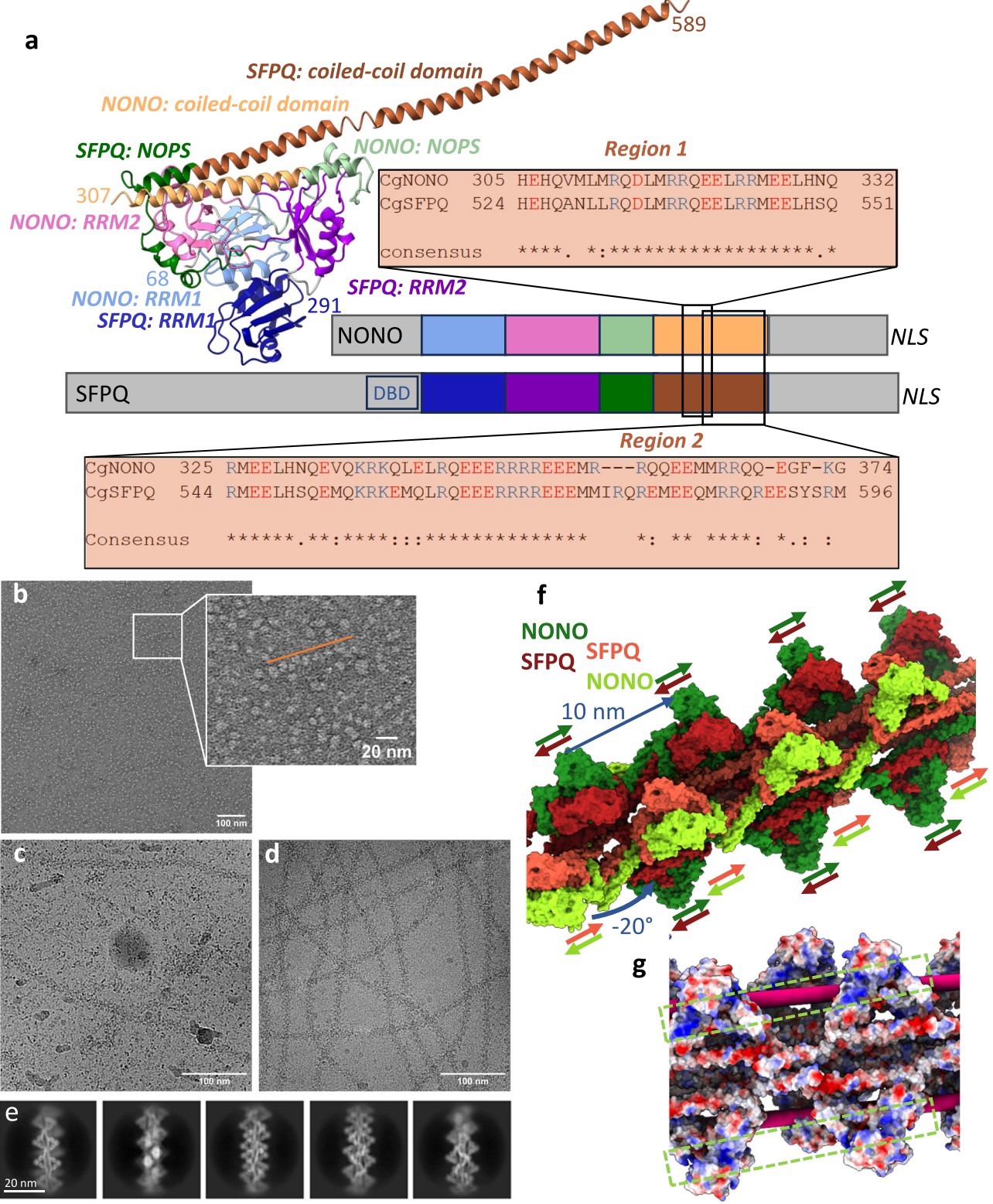

The filaments of the NONO/SFPQ-dimers consist of two intertwined strands with globular domains alternating on both sides of each strand (Fig. 1f). Within one strand the heterodimers on opposing sides are antiparallel and both intertwined strands are also antiparallel. The filaments have a left-handed twist of about -20° and a rise of 10 nm with an asymmetric unit of two heterodimers per strand (Fig. 1f; Supplementary Fig. 9).

It is well established that NONO and SFPQ preferentially form heterodimers[16,20,21,23]. The isolated dimer has a pseudo twofold symmetry with equivalent positions of SFPQ and NONO. The subunit assignment (Supplementary Fig. 10-11) showed that in the context of the double stranded filaments, SFPQ faces the opposing strand while NONO localises to the periphery (Fig. 1f) However, our data cannot exclude a low population

**Fig. 1 | Structure of NONO/SFPQ from dimers to filaments. a** Crystal structure of the human NONO/SFPQ heterodimer[22,42] (PDB: 6WMZ) showing two RNA recognition motives (RRM1, blue RRM2, pink/purple), a NonA/paraspeckle domain (NOPS, green) and a coiled-coil domain (brown). NONO is coloured in light and SFPQ in dark colours. The numbers of the N-terminal and C-terminal amino acids for NONO and SFPQ resolved in the structure are marked in the corresponding colour. A scheme of the primary sequence indicates that the conserved DBHS region (coloured the same as the structure) is flanked by disordered regions (grey) which are not conserved between NONO and SFPQ. A putative DNA binding domain (DBD) with RGG motifs is localized N-terminal of RRM1 in SFPQ. A nuclear localisation signal (NLS) can be found at the C-terminus. Sequence alignments of important regions in the coiled-coil domain between NONO and SFPQ from *Cricetulus griseus* are shown. **b** Micrographs of negatively stained NONO/SFPQ at a concentration of 50 μg/ml showing a few short chains (insert, orange line). Micrographs of vitrified

samples of NONO/SFPQ at a concentration of 0.2 mg/ml **c**, and 1 mg/ml **d**. **e** 2D class averages of filaments shown in **d**. **f** NONO/SFPQ form a left-handed helical double strand with a turn of -20° and a rise of 10 nm. SFPQ (red) faces the other strand and NONO (green) is positioned at the outside. The repeating pattern of the filaments are two structurally non-equivalent dimers (light and dark colours) on each strand. The direction of the long α-helices from N- to C-terminus is indicated by the coloured arrows, showing that the non-equivalent dimers are orientated in opposite directions along the filament. **g** Electrostatic surface representation of the model. Positively charged grooves of NONO, continuing from one head to the next, are the likely location for RNA binding (green boxes). Rigid body fitting of a RNA-bound crystal structure (PDB: 7UJ1)[19] suggests a possible RNA binding to SFPQ (pink rod) that would place the RNA between the globular domains alternating on both strands (Supplementary Fig. 17).

## Table 1 | Cryo-EM data collection, refinement and validation statistics

| EMDB ID | overall | Focused strand 1 | Focused strand 2 | Focused central strand 1 | Focused central strand 2 | Composite Structure |
|---|---|---|---|---|---|---|
| PDB ID | EMD-51413 | EMD-51417 | EMD-51418 | EMD-51438 9GLC | EMD-51439 9GLD | EMD-51471 9GNI |
| Data collection and processing | | | | | | |
| Magnification | 130,000 | | | | | |
| Voltage (kV) | 300 | | | | | |
| Electron exposure (e–/Å²) | 70 | | | | | |
| Defocus range (μm) | 0.5 to 1.4 | | | | | |
| Pixel size (Å) | 0.946 | | | | | |
| Symmetry imposed | C1 | | | | | |
| Initial particle images (no.) | 4,144,650 | | | | | |
| Final particle images (no.) | 2,974,535 | | | | | |
| Map resolution (Å) | 3.9 | 3.5 | 3.5 | 3.3 | 3.3 | — |
| FSC threshold | | | 0.143 | | | |
| Map resolution range (Å) (from conical FSCs) | 3.4-4.6 | 3.0-4.7 | 3.0-4.2 | 2.9-4.1 | 2.9-4.2 | — |
| Refinement | | | | | | |
| Initial model used (PDB code) | — | — | — | 6WMZ | 6WMZ | 6WMZ, 9GLC, 9GLD |
| Model resolution (Å) | — | — | — | 3.7 | 3.7 | 6.0 |
| FSC threshold | | | | 0.5 | 0.5 | 0.5 |
| Map sharpening B factor (Å²) | -50 | -50 | -50 | -50 | -50 | — |
| Model composition | | | | | | |
| Non-hydrogen atoms | | | | 14,423 | 14,629 | 74,499 |
| Protein residues | | | | 1728 | 1751 | 9,027 |
| Ligands | | | | 0 | 0 | 0 |
| *B* factors (Å²) | | | | | | |
| Protein | | | | 83 | 81 | 149 |
| R.m.s. deviations | | | | | | |
| Bond lengths (Å) | | | | 0.005 | 0.004 | 0.006 |
| Bond angles (°) | | | | 0.731 | 0.719 | 0.601 |
| Validation | | | | | | |
| MolProbity score | | | | 1.73 | 1.67 | 1.61 |
| Clashscore | | | | 11.34 | 11.22 | 11.06 |
| Poor rotamers (%) | | | | 0.38 | 0.00 | 0.05 |
| Ramachandran plot | | | | | | |
| Favoured (%) | | | | 97.07 | 97.46 | 97.76 |
| Allowed (%) | | | | 2.93 | 2.54 | 2.24 |
| Disallowed (%) | | | | 0.00 | 0.00 | 0.00 |

of errors within the ordered arrangement. Because of this ordered arrangement of the subunits, only SFPQ, and not NONO, contributes to the weak interactions between the strands. The cryo-EM density map suggests that the end of the long α-helix of SFPQ interacts with the NOPS domain at the loop Q461 to P464 of a SFPQ in the opposing strand (Supplementary Fig. 12). However, this contact is present only for the heterodimers on one side of each strand. For the other half, the contact to NOPS is blocked by the same α-helix of the other strand (Supplementary Fig. 12). Despite the differences in interstrand contacts, there is very little difference in the backbone (RMSD of Cα < 1 Å) between the non-equivalent SFPQ subunits (Supplementary Fig. 13).

The long α-helical regions beyond the globular domains provide a strong interaction network through different coiled-coil regions (Fig. 2). In region 1, close to the globular domain, NONO and SFPQ form a left-handed, antiparallel coiled-coil. This is stabilised by a leucin zipper and other interactions (Supplementary Figs. 4 and 14). Of notice, at one end close to the kink of SFPQ, H524 (SFPQ) forms a π-π-interaction with H330 of NONO. In the middle of region 1, R538 of SFPQ interacts with R319 of

NONO (Fig. 2). Despite their like charges, such arginine-arginine contacts frequently promote protein–protein interactions and are energetically favourable[27,28]. These conserved arginine residues from NONO and SFPQ align to each other in sequence alignments (Supplementary Fig. 4) and mutation in the *Drosophila* homologue nonA causes defects in courtship and vision[29]. A recent crystal structure of NONO/SFPQ heterodimers shows also a coiled-coil with another dimer but because the NONO construct was truncated to the globular domain, a different coiled-coil interaction is established. Instead of a contact between NONO and SFPQ, two SFPQ subunits interact and R542 (human homologue to R538 in *C. griseus*) from the different subunits contact each other[22]. In comparison with this and other crystal structures[23] we observe a more pronounced kink in the filament between the globular domain and region 1 (Supplementary Fig. 15).

Behind region 1 follows region 2 which promotes the interaction with another SFPQ subunit on the opposite side of the same strand but two heterodimers away (Fig. 2, Supplementary Fig. 9). The right-handed, anti-parallel coiled-coil is stabilised by intercalating methionine residues. Like in region 1, the centre of region 2 is formed by a cluster of charged residues

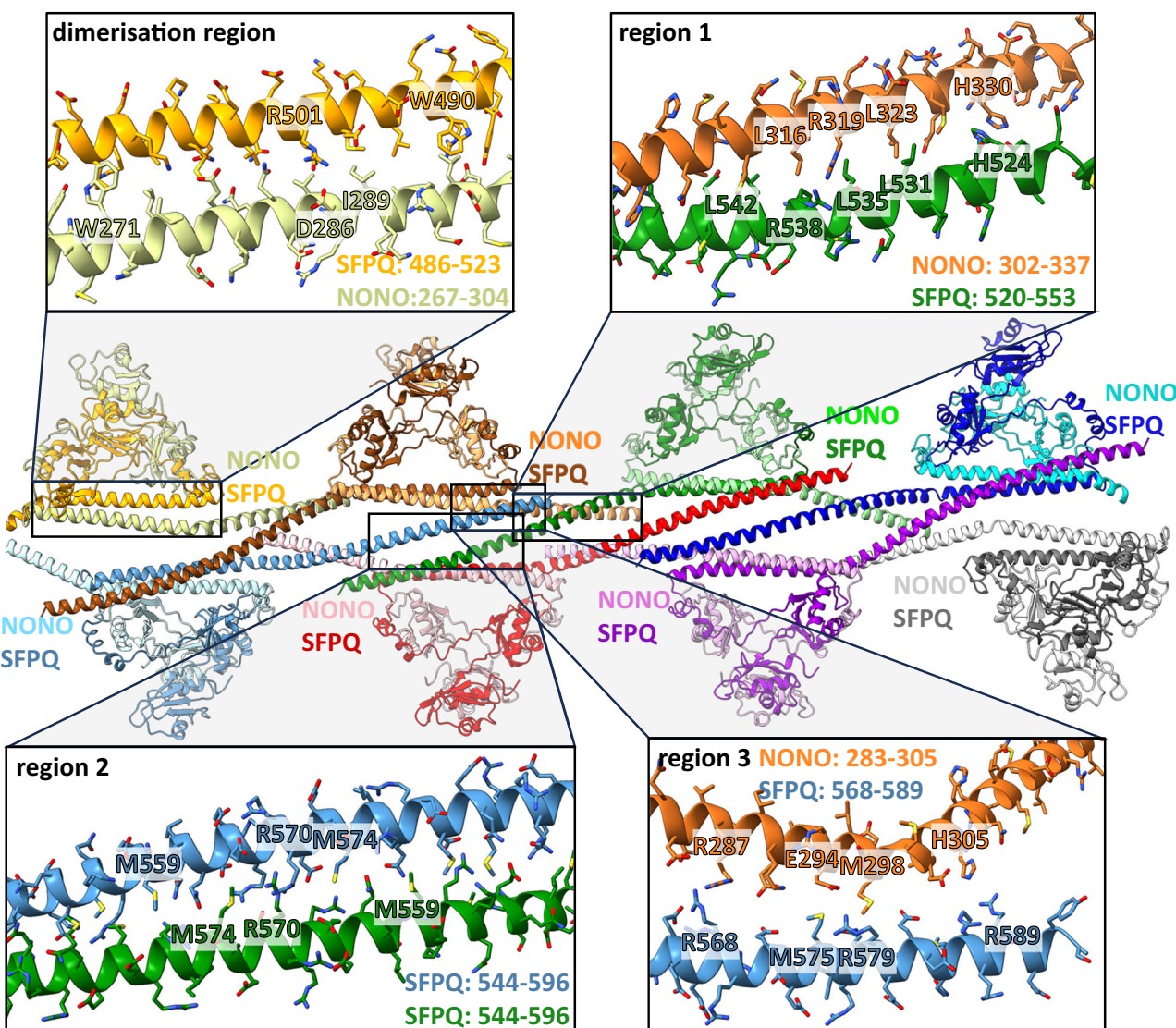

**Fig. 2 | Interactions between dimer building blocks within one strand.** The model of one of the two strands is shown in the centre for highlighting the α-helical network. Different heterodimers have different colours to explain which dimers interact with each other. NONO is shown in lighter and SFPQ in darker colours. The coiled-coil domain engages in interactions within a dimer (dimerisation region) and with other dimers within a strand (oligomerisation region). Region 1 provides interaction with the direct neighbour on the same side of the strand while regions 2 and 3 interact with dimers on the opposite side of the strand. Only one example for each interaction region is highlighted. Residue ranges for the different coiled-coils are indicated in the inserts.

(R567, R570, and E571) from the opposing SFPQs. In general, salt bridges are more relevant for interactions in region 2 in comparison to the dimerisation region and region 1 (Supplementary Fig. 14) and this region has been classified as Charged Single Alpha Helix (CSAH) region[22,30]. The R570 residues from the different subunits meet each other but take up different rotamer-conformations (Supplementary Fig. 16a). Region 2 of SFPQ also forms a coiled-coil in a crystal structure (PDB: 4WIK)[31] though in a different way as in the filaments (Supplementary Fig. 16b)[23]. The same study by Lee et al. also showed that region 2 of SFPQ is essential for paraspeckle formation in vivo[23]. Taken together, the long α-helix of SFPQ first interacts with NONO of the same heterodimer (dimerisation domain) in a right-handed coiled-coil, then with NONO of the neighbouring heterodimer in a left-handed coiled-coil (region 1), and finally with another SFPQ in a right-handed coiled-coil (region 2).

An additional short interaction site (region 3, Fig. 2) overlaps partly with region 2 in SFPQ and mediates the contact to NONO between the dimerisation domain and region 1 in the opposing heterodimer: Here, the closest contacts are R589 (SFPQ) with H305 (NONO) and M575 (SFPQ) with M298 (NONO). An arginine-arginine bridge is established between R287 of NONO and R568 of SFPQ. It is however questionable if region 3 is important for the filament stability because regions 1 and 2 were evaluated with the program PISA[32] as essential for assembly formation while region 3 only has an auxiliary role or is not significant.

In summary, we observe very specific interactions of the native full-length DBHS proteins in the filaments within these coiled-coil regions. Consequently, it seems unlikely that the filaments form just by chance and only exist in vitro. It seems more likely that the filaments serve as a molecular scaffold for gene regulation events in vivo, especially as functional relevance of oligomerisation has already been proven[23,33]. It should be mentioned that for transcription factors with sterile alpha motif (SAM)- or bric- a' -brac (BTB)-domains filament formation has also been observed[34,35]. It is thus desirable to understand how NONO-SFPQ filaments interact with RNA, DNA and other binding partners. Recently, the crystal structure of the human SFPQ homodimer in complex with RNA was determined[19]. Aligning the structure of the SFPQ-RNA complex (PDB: 7UJ1)[36] with the globular region in our filaments shows that the RNA binding site of SFPQ is close to the interstrand contact within the groove (Supplementary Fig. 17) and shielded by the globular domains of RRM1. The second RNA binding site would face the outside of the filament provided by NONO in four continuous bands of positively charged grooves (Fig. 1g).

A structure in complex with DNA has not yet been published nor has the DNA binding domain been resolved. This domain is N-terminal to the RRM1 domain on SFPQ and is therefore likely to be located on top of the globular domains, suggesting that DNA could run along the globular heads on one strand (Supplementary Fig. 17b). However, dsDNA is rigid and can only contact two or three consecutive heads in the filament before it diverges from the filament as its persistence length of 40–60 nm does not allow it to follow the twist of the filament. This could explain how the double-stranded NONO/SFPQ helix holds two ends of DNA for double-strand break repair[37], while longer continuous stretches of dsDNA cannot bind.

In conclusion, we show how NONO/SFPQ heterodimers form intertwined double-stranded helical filaments in the absence of RNA or DNA. These filaments provide a binding platform of high avidity which supports the multiple functions of DBHS proteins as a scaffold. The high flexibility could ease the interaction with the rigid dsDNA or in general allow the filament to adapt more easily to different interaction partners. Despite the high identity between NONO and SFPQ in the core they have different environments within the double stranded filaments because SFPQ faces the other strand while NONO is fully exposed to the outside. This could regulate the accessibility of RNA, DNA and protein cofactors to SFPQ and NONO.

Our filament structure differs from other oligomerisations of DBHS dimers presented before. A filamentous single strand model was proposed for NONO/PSPC1[24] where the separation of the globular domains (rise) and twist is larger in comparison to our NONO/SFPQ filament (Supplementary

Fig. 18). Oligomerisation between SFPQ subunits from different dimers in the crystal structures[22,23] differs from our structure because we see a SFPQ/NONO interaction instead of SFPQ/SFPQ in region 1 (Fig. 2). In addition, more pronounced kinks at the ends of this region 1 are present (Supplementary Fig. 15). However, similar conserved residues are involved in the coiled-coil interaction in region 1 (Supplementary Fig. 4; red bold letters), and in a crystal structure of homo SFPQ[23] interactions in region 2 are also observed (Supplementary Fig. 16b).

## Methods

### Purification of the NONO-SFPQ heterodimer

ExpiCHO cells (Gibco) were grown in suspension in Expi293 serum-free medium (Gibco). For expression, a 1 l (4 × 250 ml) suspension culture was inoculated to $0.75 \times 10^6$ cells/ml. The culture was harvested after 96 h, yielding a cell pellet of about 9 g. The cell pellet was frozen and stored at -20 °C until further use. For purification, the cells were suspended and lysed in a Teflon-in-glass homogeniser in 50 ml of lysis buffer containing 1% dodecylmaltoside (DDM, Glycon Biochemicals, Germany), 50 mM sodium phosphate buffer pH 7.5, 300 mM NaCl, 10% glycerol, 20 mM imidazole, complete EDTA-free protease inhibitor cocktail (Roche) and 1 mM phenylmethanesulfonyl fluoride at 4 °C. After 1 h of incubation on ice, 4 mg DNAse (Roche) and 4 mM $MgCl_2$ were added and incubated for another hour. After 60 min centrifugation at 30,000 g, the supernatant was applied to a 1 ml Ni-nitrilotriacetic acid (Ni-NTA) agarose column (Sigma) beforehand equilibrated with washing buffer, containing 0.05% DDM, 50 mM sodium phosphate buffer pH 7.5, 300 mM NaCl, 10% glycerol and 20 mM imidazole. A native histidine-rich motif of NONO (HHQHHHQQHH, residues 19 to 28 in CgNONO) is employed as a native binding motif for this purification step[26]. The column was washed with 40 ml washing buffer and proteins eluted in 1 ml fractions with an elution buffer, composed identically to the washing buffer, but with 300 mM imidazole. The peak fraction was applied to a 10/300 Superdex200 Increase column (Cytiva) equilibrated with SEC buffer, containing 50 mM HEPES pH 7.5, 150 mM NaCl, 5 mM EDTA and 0.03% DDM. Samples were analysed by SDS-PAGE using hand cast 15% acrylamide gels. The higher and lower main bands on a Coomassie stained gel were excised and identified as SFPQ and NONO, respectively, by mass spectrometry after tryptic digestion at the Rudolf-Virchow-Zentrum MS facility (University Würzburg), A Western blot on nitrocellulose using a PentaHis Antibody HRP conjugate (Qiagen) produced a positive result for the band identified as NONO. The main peak from the size exclusion chromatography was further concentrated with Vivaspin 100 kDa cutoff concentrators (Millipore) and 10 mM $MgCl_2$ was added before vitrification. Dynamic light scattering measurements were recorded with a DynaPro Titan (Wyatt, USA) with an acquisition time of 5 s and 30 repeats at 25 °C directly using fractions of the size exclusion chromatography.

### Electron microscopy and data analysis

**Negatively stained samples.** Copper grids with a continuous carbon film were glow discharged for 150 s in air at a pressure of $3.0 \times 10^{-1}$ Torr at medium power with a Harrick Plasma Cleaner (PDC-002) before use. The sample was diluted to 50 µg/ml with SEC buffer and incubated on the grid for 1 min. After washing with water, the grids were washed twice with 2% uranyl acetate followed by incubation with 2% uranyl acetate for 5 min and removal of excess liquid. Stained samples were imaged in a FEI Tecnai T12 electron microscope at a nominal 52,000x magnification, with an exposure of 30 electrons/Å² and a target defocus of 1 µm.

Vitrified samples: Samples were concentrated to 0.2 mg/ml or 1 mg/ml and prepared either on copper grids with holey carbon support and an additional layer of 2 nm continuous carbon (R1.2,1.3, Quantifoil) or on UltAufoil grids (R0.6/1.0, Quantifoil), respectively. The grids were glow discharged for 60 s or 150 s, respectively, in air at a pressure of $3.0 \times 10^{-1}$ Torr at medium power with a Harrick Plasma Cleaner (PDC-002) and vitrified within one hour. The samples were vitrified in a Vitrobot IV in ethane with 5 s blot time and +20 blot force. Movie data in EER format was acquired on

a Krios G3 electron microscope (ThermoFisher) equipped with a Falcon IVi direct detector and a Selectris energy filter (cryo-EM facility Würzburg). The zero-loss movies were recorded with a slit width of 5 eV at a magnification of 130,000x with a calibrated pixel size of 0.946 Å and a total exposure of 70 electrons/Å² (exposure time of 6.2 s). The target defocus range was between 0.5 and 1.4 µm for the concentrated sample. For single particle image processing, 17059 movies of the concentrated sample were recorded in EER format.

**Data analysis**. The movies were motion corrected and dose weighted in 40 fractions in a live session of the program package *CryoSparc* version 4.5[38]. All subsequent steps of image processing were also performed in *CryoSparc*, starting with the patch based CTF estimation. 4,067,909 filament segments were selected with the filament tracer. The segment images were further curated in two rounds of 2D classification followed by the selection of the best classes, retaining 3 million segment images. An initial reference volume was first determined by helical refinement from a small scouting data set and gave a resolution of 7.9 Å. Further analysis and helical refinement resulted in a map with 4.5 Å resolution while an asymmetric non-uniform refinement of the same segments provided a map with an overall resolution of 3.9 Å. An alternative strand of data analysis used the blob picker and two rounds of 2D classification were used to reduce the particle number from 4,155,725 to 3,169,157. After heterogeneous and asymmetric non-uniform refinements, a map of 3.9 Å resolution was obtained. A 3D classification in *CryoSparc* was performed into 10 classes but further refinement of subpopulations did not improve resolution. Local refinements focused on 8 or 4 dimers on one strand, excluding the RRM1 domains, and resulted in a resolution of 3.5 or 3.3 Å, respectively (Table 1). The required masks were obtained by fitting a crystal structure (see below) into the map and using the "colour zone" tool in the program *Chimera* version 1.17.3[39] to select specific regions. Both strands were refined separately and combined with the overall map to a consensus map using the combine focused maps tool of the program *Phenix* version 1.20.1[40]. Supplementary Fig. 6 gives an overview of data analysis.

**Model building**. The initial assignment of SFPQ and NONO in the map was done by unbiased model-building with *ModelAngelo*[41] without providing sequence information (Supplementary Fig. 10). The largest continuous fragments were built in the coiled-coil region. The sequences of the fragments were used in BLAST searches and identified either NONO or SFPQ homologues. This unbiased assignment was kept for the model building and confirmed by densities of specific side chains (Supplementary Fig. 11). The crystal structure of the human NONO/SFPQ heterodimer[22] (PDB: 6WMZ)[42] was fitted rigidly into the dimer locations in the filament using the program *Chimera*. The model was manually optimised using *Coot* version 0.9.8.93[43,44] in the locally-refined maps with 4 subunits for residues 148 to 340 of NONO and 367 to 594 of SFPQ which excludes the RRM1 domains. The models were refined with the realspace refinement tool of *Phenix*. An overall model of the composite map included the RRM1 domains, which were fitted as rigid bodies. Complete dimer models from the local refinement were used as templates to obtain four repeats within the filament by rigid body fitting to the composite map, visualising the overall biological assembly. Finally, the model was refined with *Phenix* realspace refinement. Map and model parameters are summarised in Table 1. Figures were prepared and subunits compared (matchmaker) in *ChimeraX*[45]. Sequence alignments were computed with *M-coffee*[46]. The isoforms X1 were taken for sequence annotation of NONO (NCBI Reference Sequence: XP_007649790.2) and SFPQ (NCBI Reference Sequence: XP_035310744.1).

**Statistics and Reproducibility**
Four independent purifications of NONO/SFPQ were performed. An initial scouting and a large data set of cryo-EM micrographs were collected from two different purifications. No further statistical analysis was conducted.

## Data availability
The atomic models for the local refinements of the central units have been deposited in the Protein Data Bank under the accession code 9GLC[47] and 9GLD[48]. A model of a composite map, better visualising the biological assembly, has the accession code 9GNI[49]. The EM map of the overall asymmetric non-uniform refinement (EMD-51413), local refinements of the strands (EMD-51417 and 51418), local refinements of the central units on each strand (EMD-51438 and 51439) and the composite map (EMD-51471) have been deposited in the Electron Microscopy Data Bank (EMDB). The uncropped SDS-PAGE image and Western blot are shown in Supplementary Fig. 19 while SEC profiles and DLS traces from Supplementary Fig. 1 are given in the Supplementary Data. All other data are available from the corresponding author on reasonable request.

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

## Acknowledgements

We thank Julia Bahner, Christian Kraft, and Michelle Endres for technical assistance. We thank Stephanie Lamer and Andreas Schlosser from the mass spectrometry facility at the Rudolf-Virchow-Zentrum. Electron Cryo Microscopy was carried out in the cryo EM-facility of the Julius-Maximilians-Universität Würzburg. The cryo EM-facility of the Julius-Maximilians-Universität Würzburg received funding from the Deutsche For-schungsgemeinschaft (DFG, German Research Foundation – Projects 359471283, 456578072, 525040890).

## Author contributions

T.R., D.G. and B.B. conceived and designed experiments. T.R., J.K., and L.S. performed experiments. T.R. and B.B. analysed the data and wrote the manuscript with corrections from J.K., L.S. and D.G.

## Funding

## Competing interests

The authors declare no competing interests.
