## [Transparent Peer Review file · Communications Biology]

The gene regulating proteins NONO and SFPQ assemble to ordered filaments

Corresponding Author: Professor Bettina Böttcher

Version 0:

Reviewer comments:

Reviewer #1

(Remarks to the Author)

Thank you for the opportunity to review the manuscript by Rasmussen and colleagues, which describes a detailed structural analysis by cryo-EM of filamentous NONO-SFPQ heterooligomers. The authors provide novel insights into the architecture and oligomerization mechanisms of proteins in the DBHS family. I particularly enjoyed the continuous narrative style, which was engaging and facilitated the reading experience. The decision not to break the manuscript into multiple subsections contributed to the clarity and flow of their arguments.

I also want to highlight the clever approach of purifying the native complex from CHO cells by utilizing a natural histidine-rich motif in NONO as a built-in affinity tag for IMAC purification, thus avoiding artificial tags and preserving the native state of the complex.

Although I am not an expert in the specific biological context the authors seek to explore, I believe I can evaluate the technical and methodological approaches and the conclusions that follow, given the authors' straightforward goal of reporting their novel structure. My assessment was facilitated by figures that illustrated complex structural findings effectively:

(1) the single-particle data are of excellent quality, achieving high global (3.9 Å) and local (up to 3.3 Å) resolutions, using a state-of-the-art microscope setup and data analysis software.

(2) their structural interpretations are thorough, clearly delineating the specific coiled-coil domains and inter-subunit interactions that stabilize the filamentous form.

(3) their identification of distinct roles for NONO and SFPQ within the filament structure provides valuable insight into their functional specialization.

(4) finally, the authors provide thoughtful hypotheses regarding the possible interactions with RNA and DNA based on the electrostatic surface analysis and structural comparisons.

While the structural data and interpretations are compelling, experimental validation of biological relevance in a physiological or in vitro context would greatly strengthen the biological significance. Additionally, biochemical experiments, such as RNA or DNA binding assays or mutagenesis (if recombinant expression is possible) of predicted binding sites, would further validate and enhance the proposed functional models. However, I understand that these experiments are outside the scope of the present manuscript, but could be valuable directions for future research.

My few minor suggestions are:

- The PISA server quantification of interfaces is always very informative, especially in the context of showing that some interfaces (coiled-coil and inter-subunit) are specific and not formed by chance (as argued by the authors on page 5, lines 140-144);

- Perhaps consider explicitly addressing the implications of filament flexibility (observed from their EM data) and its potential impact on function in more detail.

- A brief, explicit comparison with previously proposed oligomerization models could further highlight the novel structural insights presented here.

Reviewer #2

(Remarks to the Author)

The article by Rasmussen et al. reports the cryo-EM study of native and full length SFPQ-NONO purified from Chinese Human Ovary (CHO) cells taking advantage of a natural cryptic His-tag-like embedded within NONO sequence. Their study is the first to report a full-length DBHS complex structure. Indeed, in contrast to what was previously described from crystal structures and packing observation, the herein described filament are more complex than expected forming long filament

above certain concentration. This has interesting biological consequences on RNA and DNA binding.

I have some comment that should be addressed before a potential publication.

Major:

The way the authors purify the complex is very smart and elegant however, as mentioned in the introduction, DBHS proteins form obligate dimers, either as homodimers or heterodimers. DBHS proteins are also able to exchange along time as described in Lee et al. *J. Biol. Chem* 2022. Hence, when purifying a SFPQ-NONO complex, how the authors ensured that the complexes they are observing and describing are only SFPQ-NONO and not SFPQ-SFPQ or NONO-NONO dimers? Knowing that it is difficult to answer such a question due to the limited resolution and protein conservation, it might be still interesting to either i) tackle this question looking for peculiar residues that make the protein assignment unambiguous, or ii) discuss this potential discrepancy (that does not change the conclusions)

The authors make an interesting comment at the need of their manuscript (Line 163-165) that would deserve more discussion and would strengthen the output.

Minor:

In the summary, it is mentioned that coiled coil domains are responsible for dimerization between SFPQ and NONO, which is true, however the way this is mentioned seems to imply that only these domains are involved in dimerization, which is not true. Indeed, the NOPS and RRM domains are also involved. Please correct.

Reviewer #3

(Remarks to the Author)

The article by Rasmussen et al describes the structural basis for the filament formation by transcription factors of DBHS family. I have read the article with a great interest, the observation of filament formation in such a complicated way is fascinating fact. Definitely it could be published in *Communications Biology*, however the way of presenting the data and conclusions need substantial rework.

1) Authors show the concentration dependence of filament formation based on their appearance under negative staining conditions at various concentrations, however SEC profile shown at Fig S1B does not show clear signs of multimer formation but the concentration of the protein is not shown. This profile has many issues itself: it is unclear which column was used (I assumed it to be Superdex 200 Increase 10/300 mentioned in Materials and methods), calibration data is not shown, the Figure 1b is not referenced in the main text. The description of the figure is erroneous: "Size exclusion chromatography of shows an elution...". Authors conclude that the peak size does not correspond neither to multimers nor to isolated dimers, but this is misleading, since elution volume depends not solely on the mass of the particles but rather on their hydrodynamic radius, so the presence of disordered or filamentous domains could strongly increase the apparent mass. Thus, I would strongly recommend to complement SEC profiles at different concentrations by SEC-MALS, DLS or SAXS data to unambiguously judge on molecular mass of the assemblies. I would suggest that if filament formation is supposed to be concentration-dependent, authors should run SEC at various protein concentrations to prove it. SAXS, DLS or cross-linking could also be used if authors do not have sufficient amount of the protein to run SEC at higher protein concentration.

2) It is strange that filament formation was never observed in crystal structures, authors should check if the crystal contacts coincide with filament-forming interfaces and further discuss this point. Do high salt concentrations used in crystallization prevent the filament growth? This could be checked by running SEC in high salt buffer.

3) The entire Cryo-EM data processing workflow should be shown in a separate Supplementary Figure

4) Domain structure of the proteins should be shown at the Figure 1 instead of S1A. I would strongly suggest to color same domains from different proteins with different colors (For example RRM1 from NONO and SFPQ should be of different colors), and color the crystal structure at 1A accordingly. Overlay of figure parts should be avoided (coil from Fig 1a protruding into 1b).

5) The blue box in Fig 1f showing building block of the filament should not overlay the structure. I would strongly recommend to complement this figure by the sub-structures showing sequential assembly of the filament: heterodimer>tetramer>filament

6) The description of Fig 1g is unclear: "RNA binding to SFPQ (pink rod) would place the RNA in the groove between the strands."

7) Description of the interchain contacts is messy, authors should clearly subdivide description into the 1) contacts between monomers; 2) contacts between two heterodimers in the single tetramer building block; 3) contacts between tetrameric building blocks involved in multimer formation. Interfaces shown at Figure 2 should be labelled accordingly. Authors should analyze hydrogen-bonding network within the interaction interfaces.

8) The article would benefit from showing the multiple sequence alignment showing interaction interfaces for both proteins from several species (and they should necessarily include fly orthologs and third human protein of this family PSPC1) where the residues involved in heterodimerization, tetramer formation and filament formation are labelled.

9) It is worth to mention other examples of filament formation by transcription factors, for example BTB (10.1016/j.molcel.2024.05.029 10.1016/j.molcel.2024.06.010) and SAM domains (10.1038/nsb802 10.1002/prot.24645 10.1016/j.molcel.2016.06.019 10.1002/pro.2968 10.1074/jbc.M111.336115)

10) Proofreading is strongly recommended

Version 1:

Reviewer comments:

Reviewer #1

(Remarks to the Author)

The authors have addressed my comments adequately. Thank you.

Reviewer #2

(Remarks to the Author)

Dear Editors, Authors,

The modifications to the manuscript fully answer my concerns, and most probably other reviewers', comments. I fully support publication of this article in Communications Biology.

Best regards

Reviewer #3

(Remarks to the Author)

Authors addressed most of my questions, the manuscript has been substantially improved. Minor points:

- 1) Fig 1f description: “.. The repeating theme..” I think “repeating pattern” or “repeating unit” would be more appropriate wording.
- 2) Fig 1g description: authors changed the sentence about RNA binding to “A possible RNA binding to SFPQ (pink rod) would place the RNA between the globular domains alternating on both strands.” The meaning is still unclear and I understood it only after reading the description of Fig S17 (I suggest this figure to be referenced here as well), please explain here that it is about rigid body fitting of RNA-bound protein crystal structures.
- 3) Despite the authors claim that proofreading has been done, the article still suffers from awkward and unclear phrasing. I believe that consulting language-editing agency or at least using of one of the AI-powered writing tools would substantially improve its readability.

Reviewer #4

(Remarks to the Author)

I co-reviewed this manuscript with one of the reviewers who provided the listed reports. This is part of the Communications Biology initiative to facilitate training in peer review and to provide appropriate recognition for Early Career Researchers who co-review manuscripts.

We like to thank the reviewers for taking time to thoroughly evaluate our manuscript.

Reviewers' comments:

Reviewer #1 (Remarks to the Author):

We are very pleased that reviewer #1 liked our work and manuscript!

Thank you for the opportunity to review the manuscript by Rasmussen and colleagues, which describes a detailed structural analysis by cryo-EM of filamentous NONO-SFPQ heterooligomers. The authors provide novel insights into the architecture and oligomerization mechanisms of proteins in the DBHS family. I particularly enjoyed the continuous narrative style, which was engaging and facilitated the reading experience. The decision not to break the manuscript into multiple subsections contributed to the clarity and flow of their arguments.

I also want to highlight the clever approach of purifying the native complex from CHO cells by utilizing a natural histidine-rich motif in NONO as a built-in affinity tag for IMAC purification, thus avoiding artificial tags and preserving the native state of the complex.

Although I am not an expert in the specific biological context the authors seek to explore, I believe I can evaluate the technical and methodological approaches and the conclusions that follow, given the authors' straightforward goal of reporting their novel structure. My assessment was facilitated by figures that illustrated complex structural findings effectively:

(1) the single-particle data are of excellent quality, achieving high global (3.9 Å) and local (up to 3.3 Å) resolutions, using a state-of-the-art microscope setup and data analysis software.

(2) their structural interpretations are thorough, clearly delineating the specific coiled-coil domains and inter-subunit interactions that stabilize the filamentous form.

(3) their identification of distinct roles for NONO and SFPQ within the filament structure provides valuable insight into their functional specialization.

(4) finally, the authors provide thoughtful hypotheses regarding the possible interactions with RNA and DNA based on the electrostatic surface analysis and structural comparisons.

While the structural data and interpretations are compelling, experimental validation of biological relevance in a physiological or in vitro context would greatly strengthen the biological significance. Additionally, biochemical experiments, such as RNA or DNA binding assays or mutagenesis (if recombinant expression is possible) of predicted binding sites, would further validate and enhance the proposed functional models.

However, I understand that these experiments are outside the scope of the present manuscript, but could be valuable directions for future research.

We agree that the suggested experiments are valuable, and we will pursue them in the future.

My few minor suggestions are:

- The PISA server quantification of interfaces is always very informative, especially in the context of showing that some interfaces (coiled-coil and inter-subunit) are specific and not formed by chance (as argued by the authors on page 5, lines 140-144);

We analysed our structure with PISA and found that region 1 and 2 are significant for assembly formation while region 3 is not. We added following sentence: *“It is however questionable if region 3 is important for the filament stability because regions 1 and 2 were evaluated with PISA as essential for assembly formation while region 3 has only an auxiliary role or is not significant.”*

- Perhaps consider explicitly addressing the implications of filament flexibility (observed from their EM data) and its potential impact on function in more detail.

We added the sentence: *“The high flexibility could ease the interaction with the rigid dsDNA or in general adapt more easily when different interaction partners are brought together.”*

- A brief, explicit comparison with previously proposed oligomerization models could further highlight the novel structural insights presented here.

We added a final paragraph:

“Our filament structure differs from other oligomerisations of DBHS dimers presented before. A filamentous single strand model was proposed for NONO/PSPC where the separation of the globular domains (rise) and twist is larger in comparison to our NONO/SFPQ filament (Supplementary Fig. 18). Oligomerisation between SFPQ subunits from different dimers in the crystal structures differs from our structure because we see a SFPQ/NONO interaction instead of SFPQ/SFPQ in region 1 (Fig. 2) and a more pronounced kink at the ends of this region 1 are present (Supplementary Fig. 15). However, similar conserved residues are involved in the coiled-coil interaction in region 1 (Supplementary Fig. 4; red bold letters) and in a crystal structure of homo SFPQ also interactions in region 2 are seen (Supplementary Fig. 16b).”

Reviewer #2 (Remarks to the Author):

The article by Rasmussen et al. reports the cryo-EM study of native and full length SFPQ-NONO purified from Chinese Human Ovary (CHO) cells taking advantage of a natural cryptic His-tag-like embedded within NONO sequence. Their study is the first to report a

full-length DBHS complex structure. Indeed, in contrast to what was previously described from crystal structures and packing observation, the herein described filament are more complex than expected forming long filament above certain concentration. This has interesting biological consequences on RNA and DNA binding.

I have some comment that should be addressed before a potential publication.

Major:

The way the authors purify the complex is very smart and elegant however, as mentioned in the introduction, DBHS proteins form obligate dimers, either as homodimers or heterodimers. DBHS proteins are also able to exchange along time as described in Lee et al. J. Biol. Chem 2022. Hence, when purifying a SFPQ-NONO complex, how the authors ensured that the complexes they are observing and describing are only SFPQ-NONO and not SFPQ-SFPQ or NONO-NONO dimers?

Knowing that it is difficult to answer such a question due to the limited resolution and protein conservation, it might be still interesting to either i) tackle this question looking for peculiar residues that make the protein assignment unambiguous, or ii) discuss this potential discrepancy (that does not change the conclusions)

This is a very interesting point raised by the reviewer. Initially we thought that the low resolution we obtained by helical refinement might be caused by the random orientation of NONO and SFPQ within the filament. But we got better resolutions with local refinements so that the flexibility of the filament explains the low resolution in the helical refinement.

With the high similarity between NONO and SFPQ and the limited resolution of our structures, the subunit assignment is not a trivial task. We assessed the subunit assignment in two ways: An unbiased way, we described already in the original manuscript, is the automatic model building with the program Modelangelo. For this we used the highest resolution local refinements of four dimer units for both strands. The ten longest fragments of the automatic modelling (for both strands respectively) were then searched in BLAST and it was assessed if NONO or SFPQ were higher scored. Some fragments were not found by BLAST under the standard settings but all other gave a clear result. To make this assignment more transparent we added Supplementary Fig. 10 showing details. We also compared the side chain densities of the well resolved regions. A few positions allow to distinguish between NONO and SFPQ. The Supplementary Fig. 11 was added to illustrate that.

It was reported that preferentially heterodimers are formed which does not exclude that there are low populations of homodimers. However, crystal structures of the

NONO/SFPQ heterodimer were reported which would suggest that a stable and high population of the heterodimers must be achievable. From our structure we would not be able to tell if there is a low population of subunits “at the wrong” place. So, we are not able to tell if these errors are present in the filament or if the homodimers are not able to integrate into the filament. Putting it differently, our structure suggests that the majority of subunits follow the assigned order.

We adapted the following sentence to appreciate this uncertainty:

“The subunit assignment (Supplementary Fig. 10-11) showed that in the context of the double stranded filaments, the SFPQ faces the other strand while NONO is on the periphery (Fig. 1f) though our data cannot exclude a low population of errors within the ordered arrangement.”

The authors make an interesting comment at the need of their manuscript (Line 163-165) that would deserve more discussion and would strengthen the output.

We added the sentence: *“This could regulate the accessibility of RNA, DNA and protein cofactors to SFPQ and NONO.”*

Minor:

In the summary, it is mentioned that coiled coil domains are responsible for dimerization between SFPQ and NONO, which is true, however the way this is mentioned seems to imply that only these domains are involved in dimerization, which is not true. Indeed, the NOPS and RRM domains are also involved. Please correct.

The abstract was changed to “DHBS-proteins form obligate homo- and heterodimers through interactions within a globular domain and can further dynamically oligomerise through α -helical coiled-coils, which is crucial for many functions.”

Reviewer #3 (Remarks to the Author):

The article by Rasmussen et al describes the structural basis for the filament formation by transcription factors of DBHS family. I have read the article with a great interest, the observation of filament formation in such a complicated way is fascinating fact. Definitely it could be published in Communications Biology, however the way of presenting the data and conclusions need substantial rework.

1) Authors show the concentration dependence of filament formation based on their appearance under negative staining conditions at various concentrations, however SEC profile shown at Fig S1B does not show clear signs of multimer formation but the concentration of the protein is not shown.

We show now SEC profiles at 4 different concentrations. The main peak shifts to lower elution volumes at higher concentrations. In addition, a shoulder towards lower volumes is present. At one concentration peak and shoulder were closer characterised by negative stain EM and dynamic light scattering (see below).

This profile has many issues itself: it is unclear which column was used (I assumed it to be Superdex 200 Increase 10/300 mentioned in Materials and methods)

The information was added to the legend.

, calibration data is not shown,

Makers for calibration standards around the peak were added to the figure.

the Figure 1b is not referenced in the main text.

This was added.

The description of the figure is erroneous: “Size exclusion chromatography of shows an elution...”.

This was corrected.

Authors conclude that the peak size does not correspond neither to multimers nor to isolated dimers, but this is misleading, since elution volume depends not solely on the mass of the particles but rather on their hydrodynamic radius, so the presence of disordered or filamentous domains could strongly increase the apparent mass.

We agree with the reviewer and deleted this sentence.

Thus, I would strongly recommend to complement SEC profiles at different concentrations by SEC-MALS, DLS or SAXS data to unambiguously judge on molecular mass of the assemblies. I would suggest that if filament formation is supposed to be concentration-dependent, authors should run SEC at various protein concentrations to prove it. SAXS, DLS or cross-linking could also be used if authors do not have sufficient amount of the protein to run SEC at higher protein concentration.

We added SEC profiles at different concentrations showing a concentration dependent shift of the main peak. The position in the SEC won't give the correct mass, as the reviewer pointed rightly out, but the concentration dependent shift indicates a change in oligomerisation. We think that as soon as substantial filaments are formed, they won't penetrate the column, are filtered out and escape the detection. In the EM the filaments are more than 1 μm long. We closer characterized peak and shoulder at a lower concentration by negative stain EM and dynamic light scattering, shown in the updated Supplementary Fig. 1. These data suggest higher polydispersity in the shoulder than the main peak. Higher concentration samples scattered the light too strongly that DLS measurement were not possible anymore. It would be interesting to further characterise

the oligomerisation process beyond these initial experiments, but we will postpone this to future studies.

2) It is strange that filament formation was never observed in crystal structures, authors should check if the crystal contacts coincide with filament-forming interfaces and further discuss this point. Do high salt concentrations used in crystallization prevent the filament growth? This could be checked by running SEC in high salt buffer.

Most crystallographic studies used truncated proteins which allowed at most the interaction of one dimer with another dimer (where one of the coiled-coils extends beyond the globular domain). However, one study of the SFPQ homodimer (Lee et al., NAR 43, 3826 (2015)) contained long helices on both subunits of the dimer. Indeed, a linear polymerisation was observed here in one crystal form but not as our filaments. Interestingly, in region 1 similar conserved residues are involved in the coiled-coil contact between SFPQ/SFPQ as in our filament for SFPQ/NONO. However, the SFPQ/SFPQ contact in region 2 is not conserved in the homoSFPQ crystal structure but shifted by 3 helix turns (Supplementary Fig. 16b). So, it seems that hetero-contact in region 1 might be required to form the “correct” homo-contact in region 2. One could speculate that the NONO/SFPQ heterodimer is required for filament formation.

Snijder et al. indeed described a salt dependence of NONO/SFPQ in a Ni-NTA enrichment experiment (Fig 1D, RNA21, 347 (2015)). So, it could be that high salt concentrations during crystallisation prevent filament formation. Snijder et al. observed a threshold concentration of about 0.1 M NaCl which is lower than in our experiments (0.15 M NaCl). Perhaps other factors play also a role and require more investigations in the future.

3) The entire Cryo-EM data processing workflow should be shown in a separate Supplementary Figure

This is now shown in Supplementary Fig. 6

4) Domain structure of the proteins should be shown at the Figure 1 instead of S1A. I would strongly suggest to color same domains from different proteins with different colors (For example RRM1 from NONO and SPFQ should be of different colors), and color the crystal structure at 1A accordingly. Overlay of figure parts should be avoided (coil from Fig 1a protruding into 1b).

This was changed as suggested.

5) The blue box in Fig 1f showing building block of the filament should not overlay the structure. I would strongly recommend to complement this figure by the sub-structures showing sequential assembly of the filament: heterodimer>tetramer>filament

A sequential assembly of the filament cannot be implied by the structure of the filament and therefore would be misleading (see more detailed discussion under point 7).

However, we agree that our presentation may lack an easy access to the existing interactions between different dimers within the filament. We hope that an additional figure (Supplementary Fig. 9) which shows only the dimers involved in the respective interaction (in the spirit of the suggestion by the reviewer) will help to understand the structure.

6) The description of Fig 1g is unclear: “RNA binding to SFPQ (pink rod) would place the RNA in the groove between the strands. ”

This was changed to “*A possible RNA binding to SFPQ (pink rod) would place the RNA between the globular domains alternating on both strands.*”

7) Description of the interchain contacts is messy, authors should clearly subdivide description into the 1) contacts between monomers; 2) contacts between two heterodimers in the single tetramer building block; 3) contacts between tetrameric building blocks involved in multimer formation. Interfaces shown at Figure 2 should be labelled accordingly. Authors should analyze hydrogen-bonding network within the interaction interfaces.

We realised that our description of “repeating building block”, which we used in the Fig. 1 legend, is misleading. It is the question which tetramer counts as building block. The obvious combination of two dimers directly opposite on the strand, which makes it most easy to understand the built-up of the filament, have no contacts. The substantial interactions in the described regions 1-3 are each between different dimers. Consequently, no conclusion can be drawn of a possible order of assembly of the filament just from the structure, except the obligate dimers as starting point. Thus, we prefer a description of the interaction network along the most interacting long helix of SFPQ.

The messiness is inherent to the structure because not only neighbouring dimers interact and building blocks are not just stuck together like Lego blocks.

8) The article would benefit from showing the multiple sequence alignment showing interaction interfaces for both proteins from several species (and they should necessarily include fly orthologs and third human protein of this family PSPC1) where the residues involved in heterodimerization, tetramer formation and filament formation are labelled.

This was included as Supplementary Fig. 4. Beside PSPC1 and the fly ortholog, also the ortholog from *Caenorhabditis* was included that has been structurally characterized. Residues involved in dimerisation on the long helix and subsequent filament formation have been marked. In addition, interacting residue pairs are listed in Supplementary Fig. 14.

9) It is worth to mention other examples of filament formation by transcription factors, for example BTB (10.1016/j.molcel.2024.05.029 10.1016/j.molcel.2024.06.010) and

SAM domains (10.1038/nsb802 10.1002/prot.24645 10.1016/j.molcel.2016.06.019
10.1002/pro.2968 10.1074/jbc.M111.336115)

Following sentence was included: *“It should be mentioned that for transcription factors with SAM- or BTB-domains also filament formation was observed.”*

10) Proofreading is strongly recommended

Was done.

point-by-point rebuttal letter:

Reviewer #3:

Authors addressed most of my questions, the manuscript has been substantially improved. Minor points:

1) Fig 1f description: “.. The repeating theme..” I think “repeating pattern” or “repeating unit” would be more appropriate wording.

We changed the wording to “repeating pattern”

2) Fig 1g description: authors changed the sentence about RNA binding to “A possible RNA binding to SFPQ (pink rod) would place the RNA between the globular domains alternating on both strands.” The meaning is still unclear and I understood it only after reading the description of Fig S17 (I suggest this figure to be referenced here as well), please explain here that it is about rigid body fitting of RNA-bound protein crystal structures.

We changed to “Rigid body fitting of a RNA-bound crystal structure (PDB: 7UJ1) suggests a possible RNA binding to SFPQ (pink rod) that would place the RNA between the globular domains alternating on both strands (Supplementary Fig. 17)”

3) The article still has some unclear phrasing. I believe that consulting language-editing agency or at least using of one of the AI-powered writing tools would substantially improve its readability.

A native English speaker was consulted for proof reading.